# General Characteristics and Current State of Antibiotic Resistance in Pediatric Urinary Tract Infection—A Single Center Experience

**DOI:** 10.3390/antibiotics13080684

**Published:** 2024-07-24

**Authors:** Raluca Isac, Gabriela Doros, Cristiana-Alexandra Stolojanu, Ruxandra Maria Steflea, Ramona Florina Stroescu, Ioana-Cristina Olariu, Andrada-Mara Micsescu-Olah, Mihai Gafencu

**Affiliations:** 1Department of Pediatrics, “Victor Babes” University of Medicine and Pharmacy, Eftimie Murgu Square 2, 300041 Timisoara, Romania; isac.raluca@umft.ro (R.I.); steflea.ruxandra@umft.ro (R.M.S.); stroescu.ramona@umft.ro (R.F.S.); olariu.cristina@umft.ro (I.-C.O.); andrada.micsescu-olah@umft.ro (A.-M.M.-O.); mgafencu@umft.ro (M.G.); 2Emergency Hospital for Children “Louis Turcanu”, Iosif Nemoianu Street 2, 300011 Timisoara, Romania; 3Doctoral School, “Victor Babes” University of Medicine and Pharmacy, Eftimie Murgu Square 2, 300041 Timisoara, Romania; cristiana.stolojanu@umft.ro

**Keywords:** urinary tract infection, children, kidney anomalies, antibiotic resistance

## Abstract

Urinary tract infection (UTI) represents one of the most common bacterial infections in children, mainly caused by Gram-negative bacteria. Empirical antibiotic treatment is based on international and national guidelines for treating UTIs in children and is individualized with local antibiotic resistance patterns. The aim of this study is to bring a clear view of present-day particularities of UTIs in children. Methods: We analyzed 210 positive urine cultures identified in 141 pediatric patients admitted to the hospital over a 6-month period. Results: The majority of patients were females (57%) with a median age of 5 years (IQR 12), while male patients revealed a median age of 2 (IQR 7). Most patients originated from urban areas (53%). Only 18 patients (12.76%) were identified with underlying Congenital Anomalies of the Kidney and Urinary Tract (CAKUT). Escherichia Coli was the most frequent pathogen. Increased antibiotic resistance was found in commonly-used antibiotics Ampicillin and Trimethoprim/Sulfamethoxazole, and in the case of patients with CAKUT. Suitable antibiotics for treating a Gram-negative UTI are aminoglycosides, Meropenem, third-generation Cephalosporins, and Nitrofurantoin. Vancomycin upholds efficacy in treating a Gram-positive pediatric UTI. Conclusion: Periodical analysis needs to be performed in order to constantly update clinicians on uropathogenic antibiotic resistance and optimal empirical treatment options.

## 1. Introduction

Urinary tract infection (UTI) represents one of the most common bacterial infections in children [1]. Risk factors for developing UTI are represented by the presence of Congenital Anomalies of the Kidney and Urinary Tract (CAKUT), mainly urine-flow obstruction or vesicoureteral reflux (VUR), low urine output, functional constipation, insufficient genital hygiene, and female sex [2].

UTIs in children are mainly caused by Gram-negative bacteria. *Escherichia coli* is the primary leading pathogen causing UTIs in children [1,3]. The majority of UTIs in children (91–95%) develop from the ascension of the bacteria into the urinary tract from the periurethral area. The increase in susceptibility to UTI for girls is caused by anatomic factors, shorter length of urethra, and heavy colonization of the perineum with enteric bacteria [4,5]. Gender represents a factor of interest when describing UTIs in children. Gender discrepancies are due to anatomical factors, the presence of particular CAKUT in males, posterior urethral valves (PUV), and genital sexual activity infections in adolescent girls [6]. *Proteus mirabilis* and *Klebsiella pneumoniae* are more common in urinary catheterized or hospitalized children [7,8]. Bacteria causing UTIs are identified using urine culture, which requires at least 24 h of growth time and another 24 h for antibiotic susceptibility [1,3]. Prompt antibiotic treatment is essential in order to reduce the burden of this disease among children and the risk of renal scarring and further renal impairment [1,3,7]. Therefore, empirical treatment needs to be started as soon as possible, preferably in the first 24 h [1,3]. Empirical antibiotic treatment is based on international and national guidelines for treating UTIs in children and is individualized with local antibiotic resistance patterns [9,10,11].

International guidelines EAU/ESPU, as well as the American Academy of Pediatrics (AAP), suggest Amoxicillin/Clavulanic Acid, Ampicillin/Sulbactam, and as alternative, third-generation Cephalosporins or Aminoglycosides as empirical antibiotic treatment for UTIs in children [1,6,12,13]. National and local guidelines follow international references and adjust treatment to local antibiotic resistance studies [9]. Antibiotic resistance is one of the top global public health problems that leads to high mortality and morbidity rates. Misuse and overuse of antibiotics are the main causes of the development of multi-drug-resistant pathogens [14,15,16]. Results are alarming, pointing to a 42% median resistance rate of *E. coli* in 76 countries for the third-generation Cephalosporin [16]. One in every five UTIs caused by *E. coli* expressed reduced susceptibility to commonly used antibiotics—Ampicillin, Cotrimoxazole, and fluoroquinolones in 2020 [14]. Approaches to minimize antibiotic resistance are vital in order to preserve the effectiveness of current antibiotics and stop the emergence of pan-drug resistant strains [14]. Antimicrobial stewardship programs refer to suitable antibiotic use, reduction in over-the-counter antibiotic release, promotion of general recommendations regarding hand and genital hygiene, healthy alimentation, proper daily water intake, etc. [15].

In children, the presence of fever is common, UTI being the third most frequent cause; therefore, it is very important to diagnose a UTI episode and to start empirical antibiotic treatment in the first 24 h [6,12].

The aim of this study is to bring a clear view of present-day particularities of UTIs in children from Western Romania: global and CAKUT-related age/gender UTI distribution in children, UTI recurrence, diversity of uropathogens, pathogen-predictor variables (age, CAKUT status, environment, gender) correlations, general pediatric antibiotic resistance particularities, and the effect of CAKUT presence upon antibiotic resistance.

## 2. Results

During the 6-month study period, we analyzed 6461 observation charts and identified 210 positive urine cultures during the study period identified in 141 pediatric patients admitted to the Emergency Hospital for Children “Louis Turcanu” and presented an episode of UTI. The incidence of UTI out of total admissions was 2.18%. Thirty-five patients had more than one episode of UTI.

### 2.1. Baseline Characteristics of Study Population (Age, Sex, Origin, Presence/Absence of CAKUT)

The sample consisted of 57% females and 43% males, and the patient’s origin environment was predominantly urban, with 53% of patients originating from Romanian cities (58% of the females and 47% of the males).

For female patients, the median age of hospital admission was 5 years (IQR 12), while for male patients, the median age was 2 years (IQR 7) (Table 1).

A Mann–Whitney *U* test was conducted to compare the ages of patients between different sex groups since the age variable was not normally distributed. The test revealed a statistically significant difference in ages between males and females (W = 2977.5, *p* = 0.03), indicating that the age distribution differed between males and females.

A chi-squared test was conducted to examine the association between sex and the number of UTIs. The test revealed no statistically significant difference in urinary infection counts between males and females (X^2^ = 7.2662, *p* = 0.40). This suggests that the distribution of urinary infections is similar for both males and females in this sample.

Only 18 patients (12.76%) out of the entire study group were identified with the underlying presence of CAKUT by abdominal ultrasound. CAKUT types found in the present study were renal agenesis, renal dysplasia, renal hypoplasia, hydronephrosis, ureteropelvic junction obstruction, ureterovesical junction obstruction, VUR, double collecting system, and PUV.

Sex distribution between CAKUT groups was similar between sexes, while in the non-malformation group, there was female dominance (58.53%). No difference was found between the two groups (presence/absence of CAKUT) regarding the UTI recurrence (Table 2).

A Mann–Whitney *U* test was conducted to compare the ages of patients with and without kidney malformations. The test yielded a W statistic of 942 and a *p*-value of 0.47. This result suggests that there is no statistically significant difference in the age distribution between patients with kidney malformations and those without.

A chi-squared test of independence was conducted to examine the association between the number of urinary infections and kidney malformation status. The test yielded a chi-squared statistic of 9.11 and a *p*-value of 0.24. Since the *p*-value is greater than the significance level of 0.05, this result suggests that the distribution of urinary infections is similar for patients with and without kidney malformations.

The pattern of age seems to be similar between males and females, following a bimodal distribution, with the highest density of patients being 0–2 years old. A second peak in the patient density is around ages 14–15 for both males and females. Most of the patients in this study were younger than 5 years of age (Figure 1).

Regarding the age distribution of patients, we identified a larger number of older patients with UTI and underlying CAKUT (median age of 7 years (IQR 13)) in comparison with patients without malformation, who seem to be younger, with a median age of 2 years (IQR 10) (Table 2, Figure 2).

Most infections were found in two groups: toddlers (less than 1 year) in both male and female patients and children over 6 years of age, especially in female patients. The distribution was similar for children with and without reno-ureteral malformation (Figure 2).

### 2.2. Number of UTIs

The study group implied 210 positive urine cultures identified in 141 individual patients evaluated during the study period. Thirty-five patients had more than one episode of UTI during the 6-month observation period (Table 3). A total of 92% of patients had between one and two episodes of UTI, 6% had between 3 and 5 UTIs, and only three patients experienced a higher number of UTIs, 6,7, respectively, and 12 episodes. No statistical difference was found between patients with and without CAKUT regarding recurrent UTI; a quarter (25.20%) of patients without malformation had recurrent UTI, while only 22.22% of patients with CAKUT had recurrent UTI.

Most patients had no congenital anomalies of kidney and urinary tract, although some of them had multiple recurrent UTIs. Regarding sex distribution, most patients had one episode of UTI, predominantly females without malformations (Figure 3).

### 2.3. Types of Uropathogens

Two hundred and ten individual episodes of UTI were documented in our study. We classified uropathogen into four categories: Gram-negative bacteria belonging to *Enterobacteriaceae* Fam. (GNE)—79.4%, the main representatives of causative uropathogens; Gram-negative bacteria belonging to *Pseudomonas* spp. (GNP)—5.71%; Gram-positive bacteria (GP)—12.38%; and fungous—2.85% (Table 4).

A similar distribution of uropathogens between the four groups was found when comparing patients with and without malformations.

Most uropathogens belonged to the GNE group (78.89% and 73.07% in patients with CAKUT, respectively) (Table 4). Only less than one-fifth of uropathogens were Gram-positive or belonging to *Pseudomonas* spp. A particular uropathogenic distribution was found within the CAKUT group with a higher frequency of *R. ornithinolytica* (7.69%) and *Enterococcus* spp. (15.38%).

Fisher’s Exact Test was conducted to assess potential differences in the distribution of pathogens in Table 4 between the two groups of patients, with and without malformations. In the case of the pathogen *R. ornithinolytica*, the test yielded a *p*-value of 0.01, meaning that there is a significant association between malformation status and the infection with this pathogen. All other *p*-values are larger than 0.05, meaning that there are no other statistically significant associations between malformation status and the different types of uropathogens in this study. Further studies with larger sample sizes may be necessary to explore potential relationships more comprehensively.

The multinominal logistic regression model (MLM) describes the association between predictor variables such as patient age, CAKUT status, environment of origin, sex, and the response variable, represented by the UTI pathogen type, which includes four categories (Fungous, GNE, GNP, GP). The model takes into consideration potential correlations between observations because some patients have more than one sample included in the data set through the inclusion of a random effect term for each patient.

The MLM-fitted model can be described using the following equation:$$\log\left(\frac{P\left(Pathogen\;Type=Reference\right)}{P(Pathogen\;Type=k)}\right)\;=\beta_{0k}+\beta_{1k}\cdot Age+\beta_{2k}\cdot Sex+\beta_{3k}\cdot CAKUT+\beta_{4k}\cdot Environment+u_{i}$$
where *k* represents each pathogen type (GNE, GNP, GP); the reference category is the Fungous pathogen type; $\beta_{0k}$ is the intercept for pathogen type *k*; $\beta_{1k}$ and $\beta_{4k}$ are the coefficients for the predictors of Age, Sex, CAKUT status, and Environment, and $u_{i}$ is the random effect for the *i*th patient in the dataset. The logarithm of the odds ratio $\log\left(\frac{P\left(Pathogen\;Type=Reference\right)}{P(Pathogen\;Type=k)}\right)$ represents the natural logarithm of the ratio of the probability of observing outcome category *k* to the probability of observing the reference category.

The table below contains the coefficient estimates, standard errors, relative risk ratios (RRRs), Z-values, and *p*-values for each predictor in the multinomial logistic regression model. The reference category for the outcome variable Pathogen Type is Fungous, and the included predictors are Sex—Male, Environment—Urban, and CAKUT-Negative (Table 5).


**For the GNE Category, the main findings included:**
**Intercept (2.3):** The log odds of the pathogen type being GNE vs. Fungous when all predictors are at their reference levels. The RRR is approximately 9.97, meaning that the developing UTI with uropathogens from the GNE group is about 10 times more likely than Fungous in the baseline scenario, and this is statistically significant (*p*-value = 0.0002);**Age (0.01):** The RRR of 1.01 indicates a negligible increase in the likelihood of developing UTI with a bacteria GNE group with increasing age. This effect is not statistically significant (*p*-value = 0.94);**Sex—Female (−0.14):** A small negative effect. The RRR of 0.874 indicates that females are slightly less likely to have GNE compared to males, but this is not statistically significant (*p*-value = 0.882);**CAKUT-positive (−1.4):** A large negative effect. The RRR of 0.25 indicates a large reduction in the likelihood of GNE for individuals with CAKUT, though this effect is not statistically significant (*p*-value = 0.136);**Environment—Rural (−1.4):** A large negative effect. The RRR of 0.25 suggests a lower likelihood of GNE in rural environments, but it is not statistically significant (*p*-value = 0.203);**Patient effect (2.3):** A substantial positive effect. The RRR of 9.97 indicates a very high likelihood of GNE associated with the random outcome given by the patient effect. This is statistically significant (*p*-value = 0.0002). If a child is to develop UTI, it would most probably have a GNE bacteria.



**For the GNP Category, the main findings consisted of the following:**
**Intercept (1.1):** The log-odds of the pathogen type being GNP vs. Fungous when all predictors are at their reference levels. The RRR is about 3.00, which indicates a higher likelihood of developing GNP-UTI as a baseline scenario compared to fungal infection;**Age (0.08):** A positive effect. The RRR of 1.079 indicates a slight increase in the likelihood of developing a GNP-UTI with age. This effect is not statistically significant (p-value = 0.403);**Sex—Female (−1.67):** A large negative effect. The RRR of 0.19 indicates a significant reduction in the likelihood of developing GNP-UTI for females, but this effect is not statistically significant (*p*-value = 0.143);**CAKUT-positive (−1.9):** A large negative effect. The RRR of 0.15 indicates a reduction in the likelihood of GNP for individuals with CAKUT, though this effect is not statistically significant (*p*-value = 0.1749);**Environment—Rural (−1.3):** A large negative effect. The RRR of 0.273 suggests a lower likelihood of GNP in rural environments, but it is not statistically significant (*p*-value = 0.317);**Patient effect (1.1):** A positive effect. The RRR of 3.00 indicates a higher likelihood of GNP associated with the random patient effect. This is not statistically significant (*p*-value = 0.111).



**For the GP Category, the important results referred to the following:**
**Intercept (1.3):** The log odds of the pathogen type being GP vs. Fungous when all predictors are at their reference levels. The RRR is about 3.67, which is statistically significant (*p*-value = 0.049);**Age (0.08):** A positive effect. The RRR of 1.08 indicates a slight increase in the likelihood of developing GP-UTI with age. This effect is not statistically significant (*p*-value = 0.362);**Sex—Female (0.03):** A negligible effect. The RRR of 1.03 suggests little to no difference between females and males in the likelihood of GP, and this effect is not statistically significant (*p*-value = 0.973);**CAKUT-positive (−1.2):** A negative effect. The RRR of 0.30 indicates a significant reduction in the likelihood of GP for individuals with CAKUT, though this effect is not statistically significant (*p*-value = 0.266);**Environment—Rural (−2.1):** A large negative effect. The RRR of 0.12 suggests a much lower likelihood of GP in rural environments, but it is not statistically significant (*p*-value = 0.080);**Patient effect (1.3):** A positive effect. The RRR of 3.67 indicates a higher likelihood of GP associated with the random patient effect. This is significant (*p*-value = 0.049).


The multinomial logistic regression model reveals that if a patient is to have a UTI, it would probably be determined by a pathogen from the GNE group (Patient effect 2.3). Large negative results for CAKUT-GNE (*p* = 0.136) and rural-GNE (*p* = 0.2043) were found, making them more unlikely to develop UTI with GNE in CAKUT in rural patients. Also, a large negative effect was found for female GNP (*p* = 0.143), GNP-CAKUT (*p* = 0.1749), GNP-rural (0.317), GP-CAKUT (*p* = 0.266), GP-rural (*p* = 0.080), making it less likely to happen in patients with those conditions. A positive effect was found for GNE-patient effect (*p* = 0.0002), GP-patient effect (*p* = 0.049), GNP-age (*p* = 0.403), GNP-patient effect (*p* = 0.111), and GP-age (*p* = 0.362), giving insights about most probable outcome.

In Figure 4, we can visualize the relative probabilities of being in each pathogen type category, how the probability evolves with age, and how it varies with changes in environment, sex, and presence/absence of CAKUT. This model predicts that GNE pathogens (green line) have the highest probability across all predictor categories (sex, CAKUT status, age, and environment), although this probability decreases as the patients get older. The other statistically significant trend is that of the GP pathogens (purple line), which have the second highest probability in all categories; however, in contrast with the GNE pathogens, the GP pathogens incidence increases as the patients grow older. Although there are other trends observed across predictor categories, these were not found to be statistically significant due to issues discussed previously (Figure 4).

The likelihood of GNE infection was significantly higher compared to the reference category (Fungous), with an odds ratio (RRR) of 9.97 (*p* < 0.001). Age, Sex, CAKUT, and Environmental factors did not show statistically significant associations with GNE infection.

GNP infection was not significantly associated with Age, Sex, CAKUT, or Environmental factors. However, the likelihood of GNP infection was higher than that of Fungous (RRR: 3.00, *p* = 0.11).

GP infection exhibited a significantly higher likelihood compared to Fungous (RRR: 3.67, *p* = 0.05). None of the predictors, including Age, Sex, CAKUT, or Environmental factors, showed significant associations with GP infection.

### 2.4. Uropathogens’ Antibiotic Resistance

GNE was the leading causative uropathogen for developing UTIs (79.04%), with 166 positive urine cultures. Uropathogens antibiotic resistance was monitored throughout this study, and we identified high resistance rated for commonly used antibiotics (Ampicillin 78.47%, Ampicillin/Sulbactam 85.71%, Amoxicillin/Clavulanic Acid 30.19%, Ciprofloxacin 38.27%, Cefotaxime 34.65%, Gentamicin 34.38%) (Table 6).

The most common pathogen species in this study are *E. coli*, present in 106 of the total samples (50%), *Enterococcus* spp., present in 18 samples (8%), and *Klebsiella* spp., present in 37 samples (18%), together accounting for 76% of the total UTI samples. For *E. coli*, the antibiotics with the highest resistance by far are Ampicillin (72% of the samples) and Trimethoprim/Sulphamethoxazole (58%), with the other antibiotics tested showing resistance in a much smaller number of samples (Figure 5).

Higher antibiotic resistance was found for Netilmicin, Erythromycin, and Ciprofloxacin for *Enterococcus* spp., while *Klebsiella* spp. was highly resistant to Ampicillin, Gentamicin, Ciprofloxacin, Ceftazidime, and Amoxicillin/Clavulanic Acid.

Considering only the top three most common pathogen species (*E. coli*, *Enterococcus* spp., and *Klebsiella* spp.), the antibiotics with the lowest resistance are shown in Figure 5.

For *E. coli,* it seems that sensitivity is generally good, with a wide range of antibiotics showing similar results: Amikacin, Ertapenem, Imipenem, Meropenem, and Nitrofurantoin, for example, have over 94% efficacy.

GNP was responsible for 12 positive urine cultures causing UTI, showing high resistance rates (100%) for Ampicillin, Nitrofurantoin, Levofloxacin, and Trimethoprim/Sulphamethoxazole. Imipenem was resistant in 16.67% of cases, while Ciprofloxacin, Ceftazidime, and Cefepime in 25%. Meropenem, Gentamicin, and Amikacin, all with 8.33% resistance, remained effective in the treatment of UTIs in children caused by bacteria belonging to *Pseudomonas* spp.

The GP group was responsible for 21 positive urine cultures. Increased resistance (over 50%) was found for Oxacillin, Cefotaxime, Ceftazidime, and Erythromycin. Vancomycin and Linezolid remained effective against Gram-positive bacteria in children with UTI.

The analysis of antibiotic resistance patterns between CAKUT and non-CAKUT groups reveals several intriguing differences but is not statistically significant, as the CAKUT group has only 18 patients (Figure 6). The chi-squared tests for each antibiotic indicated notable discrepancies in resistance rates for specific antibiotics, highlighting potential clinical implications for treating infections in these distinct patient groups. Higher resistance rates were found in patients with UTI and CAKUT.

For the tested antibiotics, the chi-squared test yielded *p*-values, indicating, in selected cases, a substantial difference in resistance between the CAKUT and non-CAKUT groups. Pathogens’ resistance to Amikacin (*p* < 0.001), Cefepime (*p*-value < 0.015), Ertapenem (*p*-value < 0.001), Imipenem (*p*-value <0.001), Meropenem (*p*-value < 0.001), Norfloxacin (*p*-value < 0.001), Piperacillin/Tazobactam (*p* = 0.028), and Vancomycin (*p* = 0.020) made observable different resistance profiles to tested antibiotics when comparing the two groups, suggesting that CAKUT patients might exhibit different susceptibility profiles to this antibiotic compared to non-CAKUT patients, who have higher resistance rates. Conversely, other antibiotics such as Amoxicillin/Clavulanic Acid (*p* = 0.132), Ampicillin (*p* = 0.616), and Gentamicin (*p* = 0.733) did not show notable differences in resistance between the CAKUT and non-CAKUT groups.

## 3. Discussion

### 3.1. Baseline Characteristics of Study Population

UTI is among the most common bacterial infections in children [1]. The incidence varies with sex and age, UTI being more frequent in boys younger than 6 months (5.3%) and girls between 1 and 6 years (11%) [1]. In the present study, we investigated gender differences in children with UTI. A total of 57% of patients were female, while 43% were males. The median age of hospital admission was 5 (IQR 12) in females and a lower median in males of 2 years (IQR 7), which was statistically significant (*p* = 0.03). No other significant differences were found between genders regarding environment origin, UTI count, or presence/absence of CAKUT. The bimodal distribution reveals a high density of patients between 1 and 2 years of age for both sexes, but mainly young toddlers and small pediatric male patients who develop UTI. Similarly, 13–15-year-old adolescent females were identified to develop UTI with higher frequency. The EAU/ESPU guidelines reported an increased occurrence (20.3%) of UTIs in uncircumcised male infants and 5% in similar-age female infants [17]. Male infants under the age of 6 months are approximately twice as likely to have an initial UTI rather than female same-age patients. Recurrence UTI risk in both males and females is equivalent in toddlers who have experienced UTIs (32% vs. 35% respectively) [18]. Also, it is documented that young females (12–14 years old) have a 3.2-fold risk of developing UTI compared to similar-age males [19]. However, as children reach adolescent age (over 12 years), factors like sexual activity or hygiene practices influence the higher rate of UTIs in females [20]. Genger and presence/absence of CAKUT distribution showed a slightly higher occurrence of UTI for male toddlers, associating with CAKUT, and for adolescent female patients, regardless of malformation. Our results were in accordance with the literature findings [1,2,17,19,20].

UTIs in children can be the first sign of underlying CAKUT. Therefore, the European guidelines, EAU/ESPU, recommend performing renal ultrasound in the first 24 h in any febrile UTI for all children [21,22]. Abnormalities are generally found in 15% of cases, and 1–2% need urgent attention (drainage) [21]. In the present study, 12.76% of patients were identified with underlying CAKUT by abdominal ultrasound.

Patients with underlying CAKUT, particularly urine flow obstruction malformations, have a high risk of developing UTI, especially in association with VUR [23]. In the present study, the patients’ CAKUT type is implied, in order of frequency: PUV; VUR; double collecting system; vesicoureteral junction stenosis; hydronephrosis; and renal malformations.

### 3.2. Number of UTIs

In the present study, we analyzed 210 positive urine cultures identified in 141 patients. Thirty-five patients had more than one episode of UTI. Recurrent UTI, defined as more than three episodes of UTI, was identified in 7% of patients.

Several risk factors are known for recurrent UTI: constipation; poor fluid intake; avoidance of urinary voiding; UTI family history; bladder–bowel dysfunction (BBD); instrumentation of the urinary tract; and VUR [3,24,25]. In older children and adolescents, risk factors include the presence of kidney stones, sexual activity, and microflora changes due to antibiotic use [25]. Recurrent UTI may result in renal scarring and increase the risk of urosepsis, chronic urinary infection, and end-stage renal disease (ESRD) [1,2,24,25]. CAKUT by itself is not necessarily a risk factor for recurrent UTI [25]. Some particular types of CAKUT increase the risk of recurrent UTI, which implies obstruction in the urine outflow or reflux of infected urine to the upper urinary tract [24]. In the present study, we only evaluated CAKUT as a predisposing factor for recurrent UTI. No statistical difference was found between CAKUT and no-malformation patients regarding recurrent UTI; 25.20% of no-malformation patients had recurrent UTI, while only 22.22% of patients with CAKUT had recurrent UTI.

### 3.3. Types of Uropathogens

Urinary tract infection in children is caused mainly by Gram-negative bacteria belonging to *Enterobacteriaceae* Fam., most frequently *E. coli*, followed by *Klebsiella* spp. and *Enterobacter* spp. [8,9,10,26]. Gram-positive bacteria rarely cause urinary tract colonization [10]. In the present study, *E. coli* was responsible for 50.47% of all UTIs, while other studies report higher prevalence among uropathogens, ranging from 50 to 90% worldwide, 60–90% in Europe, and 56.43% in Central Romania [8,27]. A low prevalence of *E. coli* can raise concerns regarding other more aggressive, antibiotic-resistant, and opportunistic uropathogens.

A similar distribution of uropathogens between the four groups was found also in the present study. We also compared uropathogen distribution in patients with and without malformations. Overall, most uropathogens belonged to the GNE group, analogous to the literature findings [8,9]. A higher incidence of opportunistic bacteria was cited for the etiology of UTI in patients with CAKUT [28].

In CAKUT patients, we found a slightly different etiology of uropathogens. *E. coli* persisted as the main uropathogen causing UTIs in children, regardless of their malformation status, but then with an insignificant difference of 51.6% of UTIs caused by *E. Coli* in no-malformation patients and 42.3% in patients with CAKUT (*p* = 0.41). No statistically significant difference was found between CAKUT and patients without malformation. A particular uropathogenic distribution was found within the CAKUT group with a higher frequency of *R. ornithinolytica* (7.69%) and *Enterococcus* spp. (15.38%) (*p*-value 0.01 on Fisher test), most probably due to urinary catheterization or invasive maneuver exposure and repeated hospitalizations.

The multinomial logistic regression model applied in the present study reveals that if a patient is to have a UTI, it would probably be determined by a pathogen from the GNE group (Patient effect 2.3). A large negative result for the associations CAKUT-GNE and rural-GNE was found, leaving it unlikely to develop UTI with GNE in CAKUT and/or rural patients. Also, a large negative effect was found for the associations to female GNP, GNP-CAKUT, GNP-rural, GP-CAKUT, and GP-rural, making it less likely to develop UTI in a patient in those conditions. A positive effect was found for the GNE-patient effect (*p* = 0.0002) and GP = patient effect (*p* = 0.049), GNP-age (*p* = 0.403), GNP-patient effect (*p* = 0.111), and GP-age (*p* = 0.362), giving insights about most probable outcomes. The random effect term is a significant predictor for both GNE and GP categories, indicating variability in pathogen-type likelihoods across different patients. Age, Sex—Female, CAKUT-positive, and Environment—Rural are not statistically significant predictors in this model, meaning that they do not significantly affect the likelihood of being classified into the GNE, GNP, or GP categories compared to the reference category Fungous.

The likelihood of GNE infection was significantly higher compared to the reference category (Fungous), with an odds ratio (RRR) of 9.97 (*p* < 0.001). GNP infection was not significantly associated with Age, Sex, CAKUT, or Environmental factors. However, the likelihood of GNP infection was higher than that of Fungous (RRR: 3.00, *p* = 0.11). GP infection exhibited a significantly higher likelihood compared to Fungous (RRR: 3.67, *p* = 0.05). The random effect term is a significant predictor for both GNE and GP categories, indicating variability in pathogen-type likelihoods across different patients.

Other studies treasure similar results with a high proportion of Gram-negative bacteria, especially *E. coli,* causing UTIs in children. Racial differences lead to different risk factors, with Hispanic and white children being at two to four-fold risk of developing UTI than black children, especially in female patients [29,30,31]. Other studies revealed higher incidence rates of *E. coli* (up to 70–80% of all UTIs), and their frequency depending on study group selection, hospital admission, and prevalence of CAKUT within the study group [29,32]. Other frequent uropathogens were *E. aerogenes*, *K. Pneumoniae*, *P. mirabilis, Citrobacter* spp., *P. Aeruginosa*, *Enterococcus* spp., and *Serratia* spp. [5,32].

### 3.4. Uropathogens’ Antibiotic Resistance

European guideline EAU/ESPU endorses empiric wide-spectrum antibiotic use for UTI treatment via oral administration in children older than 2 years of age and in the first 24–48 h since symptoms debut and adapting treatment 48–72 h later, based on urine culture and antibiotic resistance [1]. Empirical antibiotic treatment is crucial in reducing the risk of kidney scarring, sepsis, and renal impairment [33,34]. Young age and underlying genitourinary malformations are known to increase the risk of resistant microorganisms and clarify the increased rate of insufficient empirical treatment [35]. Determining local resistance patterns based on regular antibiotic resistance studies is crucial for the positive outcome of empirical treatment [33,34].

European EAU/ESPU UTI treatment guidelines recommend empirical treatment with Amoxicillin/Clavulanic Acid, Ampicillin/Sulbactam, or third-generation Cephalosporins or aminoglycosides as alternative [6,12,13]. The safety of Flu0roquinolone use in the pediatric population has been recurrently debated [36,37,38]. Ciprofloxacin, Levofloxacin, Moxifloxacin, and Ofloxacin are approved for pediatric use with specific indications (complicated UTI/pyelonephritis, neonatal bacterial infections, multidrug-resistant Gram-negative organisms) [36,39]. This particular class of antibiotics is not routinely recommended in children as a first-line treatment due to adverse reactions: musculoskeletal impairment, neuropathy, memory loss, hearing, taste, smell, vision impairment, QTc prolongation, etc.) [37,40]. Musculoskeletal adverse effects caused by Ciprofloxacin have been reversible, indicative of safe use in pediatric patients [41].

Bacteria from the GNE group were the main leading cause of UTIs in children (79.04%). Antibiotic resistance was found for commonly used antibiotics. High resistance rates (over 50%) were found for Ampicillin 78.47%, Ampicillin/Sulbactam 85.71%, Amoxicillin 100%, Norfloxacin 50%, Trimethoprim/Sulphamethoxazole 61.49%, and Colistin 50%. Intermediary resistance rates (30–50%) were found for Amoxicillin/Clavulanic Acid 30.19%, Cefotaxime 34.65%, Gentamicin 34.38%, Ciprofloxacin 38.27%, and Nalidixic Acid 33.33%. Low resistance rates and, therefore, indications as alternative therapies were Amikacin, Imipenem, Meropenem, Levofloxacin, Piperacillin/Tazobactam, and the third-generation Cephalosporins. Our findings were congruent with national findings, with slightly lower resistance rates for Amoxicillin/Clavulanic acid and Gentamicin and significantly better than resistance rates in patients with CAKUT and UTI where the second-generation Cephalosporins are highly resistant, along with aminopenicillins [9,10]. Concerning resistance was found for Ciprofloxacin (38.27%), an antibiotic that is not commonly prescribed to children. A possible explanation would be the intensive use of this particular antibiotic in adults or animals generating resistant GNE strains.

Suitable antibiotics in treating UTI caused by *Enterobacteriaceae* Fam. are aminoglycosides, Meropenem, the third-generation Cephalosporins, and Nitrofurantoin.

Bacteria from the GNP group were responsible for 5.71% of UTIs in the present study, similar to other national studies’ findings [9,10,42]. A higher prevalence (9.23%) of *P. aeruginosa,* the most representative strain, is found in groups of patients with underlying CAKUT who develop UTI, while in the general pediatric population, the occurrence is between 1.5 and 2% [9,10,43]. Concern exists worldwide regarding the emergence of carbapenem-resistant strains, which represents a major issue in public health [44]. Extremely high resistance rates (up to 100%) were found in our study for Ampicillin, Levofloxacin, Nitrofurantoin, and Trimethoprim/Sulphamethoxazole. Antibiotics good for use in the case of GNP-UTI remained the third-generation Cephalosporins, fluoroquinolones, aminoglycosides, Imipenem, and Meropenem.

Gram-positive bacteria causing UTIs in children are responsible for 12.38% of urinary infection episodes. Similar to the GNP group, Gram-positive bacteria have a higher occurrence in patients with CAKUT who develop UTI, ranging from 18.07% to 29.6% [9,28]. High resistance rates were found in our study for Oxacillin (62.5%), the third-generation Cephalosporins (50%), and Erythromycin (82.61%), numbers comparable with the literature findings for resistance rates: Erythromycin from 55% to 71%, Clarithromycin 54%, Gentamicin 97.5%, Ampicillin from 31.5% to 61%, Tetracycline 56%, Ciprofloxacin 100%, and Cefuroxime 98.4% [9,28,43]. Vancomycin and Linezolid remained effective against Gram-positive bacteria causing UTIs in children.

Results from comparing the resistance patterns for patients with and without CAKUT who developed UTI do not reveal impressive statistical differences, although several findings may be of interest. Higher antibiotic resistance rates are found in patients with CAKUT and UTI for Amikacin (*p* < 0.001), Cefepime (*p* < 0.015), Ertapenem (*p*-value < 0.001), Imipenem (*p*-value < 0.001), and Meropenem (*p*-value < 0.001), which also demonstrated significant differences in resistance rates, emphasizing the need for careful consideration when prescribing these antibiotics to CAKUT patients.

Other antibiotics, such as Amoxicillin/Clavulanic Acid, Ampicillin, and Gentamicin, did not show significant differences in resistance patterns between the CAKUT and non-CAKUT groups. This suggests that these antibiotics might be equally effective (or ineffective) in both patient populations, and their use can be considered without additional bias toward one group.

A small number of studies compare antibiotic resistance in patients with CAKUT, and yet, our findings bring precious information about local antibiotic resistance patterns with CAKUT particularities [45,46]. The strength of our study is that it is one of few that inquire about the differences in antibiotic resistance between patients with and without malformations and raise awareness about increased antibiotic resistance.

### 3.5. Limitations

The main limitation of the present study is the small sample size of the study group, especially for patients with CAKUT. Also, the limited 6-month study period and single center experience from a specific region of the country represent weaknesses of the present study. Future studies on larger study populations are desirable. The present study analyzed only a few data intended to identify risk factors for recurrent UTIs.

## 4. Materials and Methods

### 4.1. Study Design

Over a six-month period (26 December 2022–28 June 2023), we retrospectively analyzed 210 positive urine cultures causing UTI identified in 141 pediatric patients admitted to the Emergency Hospital for Children “Louis Turcanu”, Timisoara. The hospital is the biggest in Western Romania, the regional hospital with a catchment area of one million people, and is responsible for the pediatric population of about one-eighth of the country`s area, with almost 40,000 emergency room presentations/year and 17,000 hospital admissions/year.

Inclusion criteria in this study were represented by the following criteria:Age between 0 months and 18 years;Positive urine culture during hospital admission.

Exclusion criteria were as follows: age over 19 years; outpatients.

This study was approved by the Ethical Committee of Emergency Hospital for Children “Louis Turcanu”, Timisoara, decision number 93/2022 and registry number 16917/20, December 2022.

### 4.2. Data Collection

The following clinical and demographical data were collected for each patient: age; sex; origin environment; type of CAKUT; type of uropathogen; and uropathogenic antibiotic resistance.

Positive urine culture was defined by more than 1000 CFU/mL when collected by bladder catheterization; values between 10,000 and 100,000 CFU/mL were collected from the middle of the urinary stream in patients with UTI symptoms or higher than 100,000 CFU/mL in patients with or without symptoms. Clinical manifestations of UTI included fever, nausea, vomiting, dysuria, pollakiuria, abdominal pain, and lumbar pain. Data collected included both upper and lower UTIs.

CAKUT diagnosis was performed by abdominal ultrasonography, and selected cases required computer tomography and/or voiding cysto-urethrogram. CAKUT diagnosis was either prior to UTI or established with UTI event. Abdominal ultrasound was executed by a trained pediatric nephrologist or a radiologist. CAKUT of interest in the present study were renal malformation (agenesis, dysplasia, hypoplasia, fusion anomalies), urine-flow obstruction malformations (hydronephrosis, ureteropelvic junction obstruction, ureterovesical junction obstruction), VUR, ureteral duplication, and posterior urethral valves.

### 4.3. Urine Sampling

The urine samples were collected by bladder catheterization from the middle of the urinary stream in wide-mouth sterile containers or using collector bags in pediatric patients under the age of 3.

Uropathogens were classified into four categories as follows:**Gram-negative bacteria (GNE)** belonging to the *Enterobacteriaceae* family, represented by *E. coli*, *Klebsiella* spp. (*K. pneumoniae*, *K. oxytoca*), *Enterobacter* spp. (*E. aerogenes*, *E. cloacae*), *Proteus* spp. (*P. mirabilis*), *Citrobacter* spp. (*C. amalonaticus*), *Serratia* spp. (*S. marcescens*), *Raoultella* spp. (*R. ornithinolytica*), and *Morganella* spp. (*M. morganii*);**Gram-negative bacteria (GNP)** belonging to *Pseudomonas* spp. (represented by *P. aeruginosa*, *P. fluorescens*), *Stenotrophomonas* spp. (*S. maltophilia*), and *Ralstonia* spp. (*R. pickettii*);**Gram-positive bacteria (GP):** *Staphylococcus aureus*, *Streptococcus* spp. (*S. agalactiae*), *Enterococcus* spp. (*E. faecium*), and *Kocuria kristinae*.**Fungous:** *Candida albicans*, *Candida tropicalis*.

### 4.4. Antimicrobial Susceptibility

In regard to the urine cultures and antibiotic susceptibility, standard protocol and methodology were followed.

The bacterial strains isolated from each sample were identified by biochemical testing on VITEK^®^ 2 Compact 15, VITEK 2 G-positive (GP), and VITEK 2 G-negative (GN) identification cards (bioMérieux, Marcy l’Etoile, France), following the manufacturer’s guidelines. For fungous cultures, direct microscopy and histopathology were used to identify pathogens. Antibiotic sensibility was tested using the Kirby–Bauer disk diffusion method, according to CLSI (Clinical and Laboratory Standards Institute) guidelines. The bacterial suspensions were adjusted to 0.5McFarland standard solution and swabbed on the surface of Muller–Hinton agar (MHA) with a stick of 0.01 mL with a standard calibrated loop and incubated at 35–37 Celsius degrees for 18–24 h. The isolates were considered resistant, intermediate, or susceptible to one antibiotic based on the diameter of the halo that appeared after the incubation period. The classification into the three bacterial groups (resistant, intermediate, or susceptible) regarding antibiotic susceptibility was carried out according to the interpretation tables of CLSI guidelines [47]. All the samples were examined regularly in the hospital’s standard accredited ISO 15189 laboratory [48]. Positive urine cultures for uropathogens belonging to Gram-negative *Enterobacteriaceae* Fam. (GNE group) were tested for thirty-three antibiotics, Gram-negative bacteria belonging to *Pseudomonas* spp., *Stenotrophomonas* spp., and *Ralstonia* spp. (GNP group) were tested for seventeen antibiotics, while Gram-positive strains (GP group) were tested for twenty-eight antibiotics.

Antibiotic resistance was categorized into three categories: high—over 50% of tested strains showed resistance; intermediary—30 to 50% of tested strains exhibited resistance; and low antibiotic resistance—less than 30% of tested strains were resistant.

### 4.5. Statistical Analysis

All data were meticulously recorded in a secure computerized database using Microsoft Excel^®^ version 2312 (Build 17126.20132); this version was released on 9 January 2024.

Statistical analysis was conducted using the software package R Version 4.2.3 [48].

Descriptive statistics were used for numerical variables as medians and interquartile ranges (IQR), whereas for categorical variables, frequency as percentages (%) and/or counts (n) were used.

The chi-squared test statistic (X-squared) and Fisher’s Exact Test were used for measuring associations where data were of a categorical nature and in specific cases when the counts in certain categories were very low. Where the data were numerical, a Mann–Whitney *U* test was used to assess whether two independent samples came from populations with the same distribution.

Throughout all statistical tests carried out, the significance level was set to 0.05. As such, a *p*-value obtained from a statistical test that is less than 0.05 indicates a statistically significant association between the variables in the test, suggesting that the observed distribution is unlikely to have occurred by chance alone.

A model that accounts for the dependent nature of the data and estimates the magnitude of the association between a chosen response variable and various predictor variables is used in the analysis of pathogen data. As such, a particular case of a Generalized Linear Mixed Effects Model (GLMM) in the form of a multinomial logistic regression model was applied using the library net) from the software package R Version 4.2.3 [48,49].

## 5. Conclusions

*E. Coli*, the most frequent uropathogen causing UTIs in children with or without malformations, was found to have a low prevalence in our study group, 50.47%, the lowest in Europe. Increased antibiotic resistance was found in commonly used antibiotics Ampicillin, Trimethoprim/Sulfamethoxazole, and Ciprofloxacin. Suitable antibiotics in treating UTI caused by *Enterobacteriaceae* Fam. are aminoglycosides, Meropenem, the third-generation Cephalosporins, and Nitrofurantoin.

High resistance rates were found for Levofloxacin and Nitrofurantoin in treating UTI with *Pseudomonas* spp. Ideal antibiotic treatment for UTI caused by *Pseudomonas* spp. remains the third-generation Cephalosporins, Ciprofloxacin, aminoglycosides, Imipenem, and Meropenem.

Gram-positive bacteria presented high resistance toward Oxacillin, Erythromycin, and Clindamycin. Vancomycin and Linezolid remained effective against Gram-positive bacteria causing UTIs in children.

Increased antibiotic resistance rates are found in patients with CAKUT and UTI for Amikacin, Cefepime, Ertapenem, carbapenems, Norfloxacin, and Vancomycin. The presence of underlying reno-ureteral malformation increases the risk of antibiotic resistance for commonly used antibiotics.

Periodical analysis is needed in order to constantly update clinicians on uropathogenic antibiotic resistance patterns and optimal empirical treatment options.

### Future Perspectives

Due to the small sample size of the group, further studies should be conducted, including patients from multiple centers, in order to have a wider view regarding the antibiotic resistance of microorganisms.

Collaboration with general practitioners and ambulatory care pediatricians brings valuable data on the current outpatient situation and, altogether, awareness about antibiotic resistance rates and initiation of prompt measures to keep control over this matter.

## Figures and Tables

**Figure 1 antibiotics-13-00684-f001:**
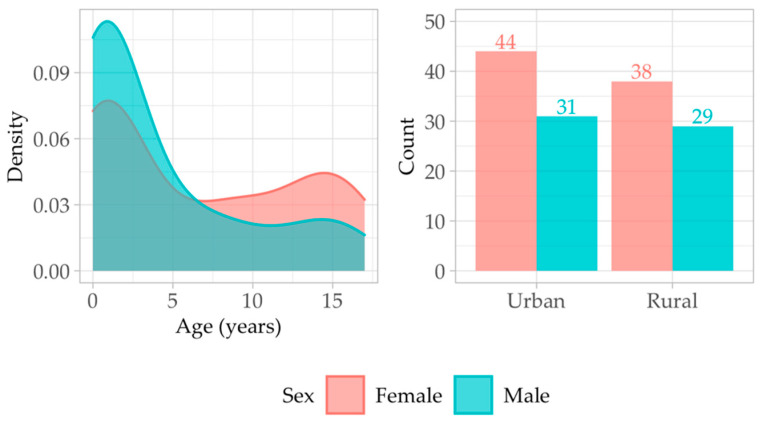
Age-sex distribution of patients (**left**); sex-environment distribution (**right**). Red square—female patients; blue square—male patients.

**Figure 2 antibiotics-13-00684-f002:**
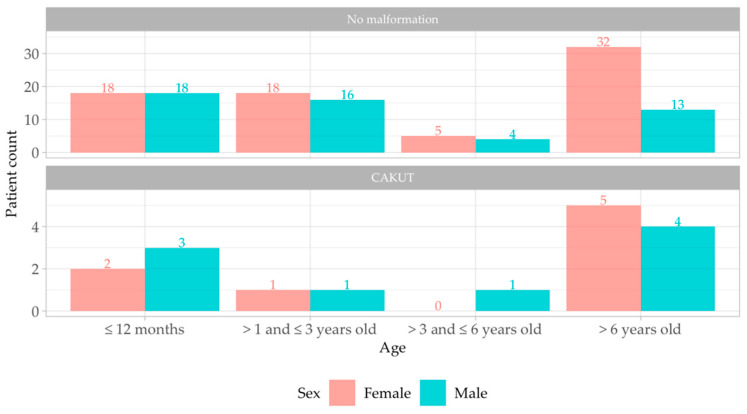
Patient age distribution with presence/absence of malformation. Red square—female patients; blue square—male patients; CAKUT—Congenital Anomalies of Kidney and Urinary Tract.

**Figure 3 antibiotics-13-00684-f003:**
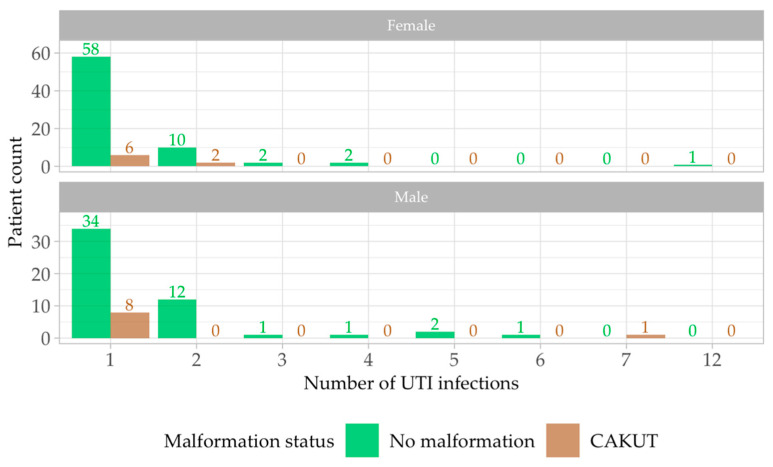
Patient sex distribution with number of UTIs and presence/absence of malformation. Green square—no malformation. Brown square—CAKUT.

**Figure 4 antibiotics-13-00684-f004:**
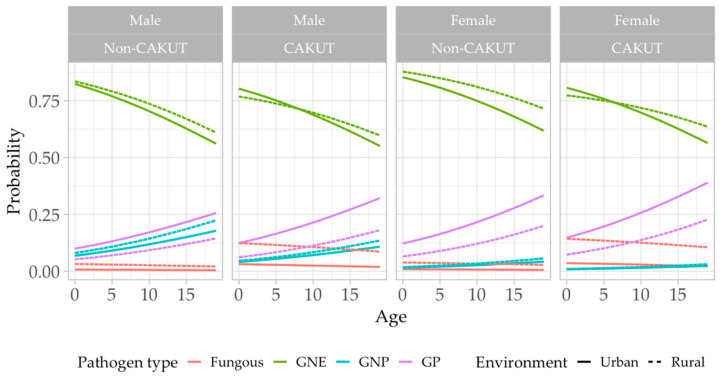
UTI pathogens model: Predictive probability plot. GNE—Gram-negative bacteria belonging to *Enterobacteriaceae* Fam.; GNP—Gram-negative bacteria belonging to *Pseudomonaceae* spp.; GP—Gram-positive bacteria. Red line—Fungous; green line—GNE pathogen; blue line—GNP pathogen; purple line—GP pathogen. All continuous lines refer to urban origin environment, and all dotted lines refer to rural origin environment.

**Figure 5 antibiotics-13-00684-f005:**
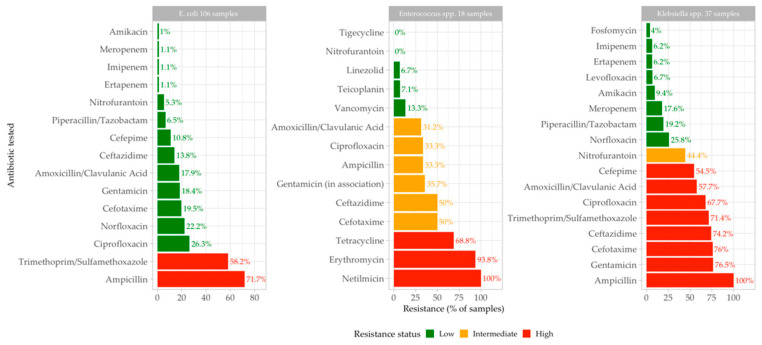
Antibiotics with highest and lowest resistance (%) for *E. coli*, *Enterococcus* spp. and *Klebsiella* spp. in our study group. Antibiotic resistance was categorized into three categories: high—over 50% of tested strains showed resistance; intermediary—30 to 50% of tested strains exhibited resistance; and low antibiotic resistance—less than 30% of tested strains were resistant.

**Figure 6 antibiotics-13-00684-f006:**
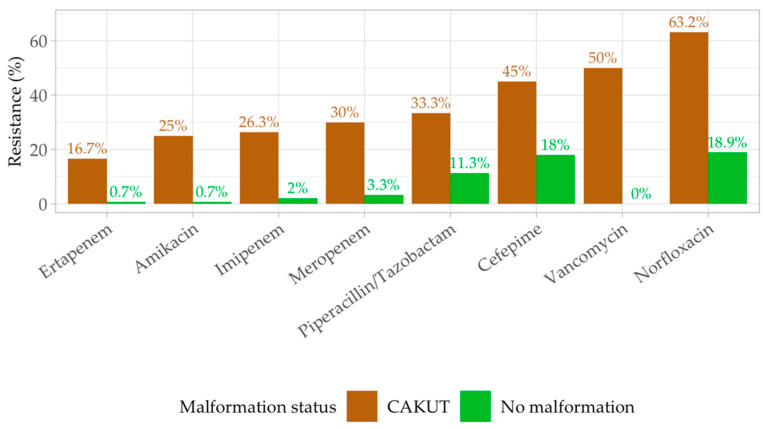
Antibiotic resistance comparison in patients with and without malformation. Green square—no malformation. Brown square-CAKUT.

**Table 1 antibiotics-13-00684-t001:** Study group characteristics, sex distribution. Statistical analysis between sex groups.

	Female			Male			*p*-Value
	N	Median	IQR	N	Median	IQR	
Age (years)	81	5	12	60	2	7	0.03 *†
Origin environment	81			60			
Urban	47	58.02%		28	46.66%		
Rural	34	41.97%		32	53.33%		
UTI count	81	1	0	60	1	1	0.40 ‡
Presence/Absence of CAKUT	81			60			
No malformation	72	88.88%		51	85%		
CAKUT	9	11.11%		9	15%		

UTI—Urinary tract infection; CAKUT—Congenital Anomalies of Kidney and Urinary Tract; IQR—interquartile ranges; *—*p*-value less than 0.05 is considered statistically significant; †—*p*-value obtained using a Mann–Whitney *U* test; ‡—*p*-value obtained using a chi-squared test.

**Table 2 antibiotics-13-00684-t002:** Study group characteristics, distribution of patients in accordance with presence/absence of malformation. Statistical analysis between groups.

	No Malformation			CAKUT			*p*-Value
	N	Median	IQR	N	Median	IQR	
Age (years)	123	2	10	18	7	13	0.47 †
Sex	123			18			
Female	72	58.53%		9	50%		
Male	51	41.46%		9	50%		
Origin environment	123			18			
Urban	63	51.31%		12	66.66%		
Rural	60	48.78%		6	33.33%		
UTI count	123	1	1	18	1	0	0.24 ‡

UTI—Urinary tract infection; CAKUT—Congenital Anomalies of Kidney and Urinary Tract; IQR—interquartile ranges; †—*p*-value obtained using a Mann–Whitney *U* test; ‡—*p*-value obtained using a chi-squared test.

**Table 3 antibiotics-13-00684-t003:** UTI count, presence/absence of malformation, sex, and origin distribution.

UTI Count	1	2	3	4	5	6	7	12
Presence/absence of CAKUT No malformation CAKUT	106	24	3	3	2	1	1	1
	92 (86.79%)	21 (87.5%)	3	3	2	1	-	1
	14 (13.20%)	3 (12.5%)	-	-	-	-	1	-
Sex Female Male	106	24	3	3	2	1	1	1
	64 (60.37%)	12	2	2	-	-	-	1
	42 (39.62%)	12	1	1	2	1	1	-
Origin environment Urban Rural	106	24	3	3	2	1	1	1
	60 (56.60%)	11	2	1	1	-	-	-
	46 (43.39%)	13	1	2	1	1	1	1

CAKUT—Congenital Anomalies of Kidney and Urinary Tract; UTI—urinary tract infection.

**Table 4 antibiotics-13-00684-t004:** Uropathogen group distribution to GNE, GNP, GP, and fungous in patients with and without malformation.

	No Malformation		CAKUT		Fisher’s Exact Test *p*-Value
	N = 184	%	N = 26	%	
**Fungous:** *Candida* spp.	4	2.17%	2	7.69%	0.16
**GNE**	147	79.89%	19	73.07%	
*E. Coli*	95	51.63%	11	42.3%	0.41
*Klebsiella* spp.	33	22.44%	4	15.38%	1
*Proteus* spp.	2	1.08%	-	-	1
*Enterobacter* spp.	12	6.52%	1	3.84%	1
*M. morganii*	4	2.17%	-	-	1
*R. ornithinolytica*	-	-	2	7.69%	0.01 *
*Citrobacter* spp.	1	0.54%	1	3.84%	0.23
**GNP**	11	5.97%	1	3.84%	
*Pseudomonas* spp.	9	4.89%	1	3.84%	1
*S. maltophilia*	2	1.08%	-	-	1
**GP**	22	11.95%	4	15.38%	
*Enterococcus* spp.	14	7.6%	4	15.38%	0.25
*Staphylococcus* spp.	7	3.8%	-	-	0.6
*K. kristinae*	1	0.54%	-	-	1

CAKUT—Congenital Anomalies of Kidney and Urinary Tract; GNE—Gram-negative bacteria belonging to *Enterobacteriaceae* spp.; GNP—Gram-negative bacteria belonging to *Pseudomonaceae* spp.; GP—Gram-positive bacteria; * *p*-value < 0.05, statistically significant. In Bold—uropathogen group.

**Table 5 antibiotics-13-00684-t005:** Association between predictor variables (age, CAKUT status, environment, sex) and response variable (Uropathogen type) by multinominal logistic regression model (MLM).

Category Predictor	Estimate	Standard Error	Relative Risk Ratio	Z-Value	*p*-Value
**GNE**					
GNE Intercept	2.30	0.62	9.97	3.71	0.0002 *
GNE Age	0.01	0.08	1.01	0.07	0.941
GNE Sex—Female	−0.14	0.91	0.87	−0.15	0.882
GNE CAKUT-Positive	−1.40	0.94	0.25	−1.49	0.136
GNE Environment—Rural	−1.40	1.10	0.25	−1.27	0.203
GNE Patient Effect	2.30	0.62	9.97	3.71	0.0002 *
**GNP**					
GNP Intercept	1.10	0.69	3.00	1.59	0.111
GNP Age	0.08	0.09	1.08	0.84	0.403
GNP Sex—Female	−1.67	1.14	0.19	−1.46	0.143
GNP CAKUT-Positive	−1.90	1.40	0.15	−1.36	0.175
GNP Environment—Rural	−1.30	1.30	0.27	−1.00	0.317
GNP Patient Effect	1.10	0.69	3.00	1.59	0.111
**GP**					
GP Intercept	1.30	0.66	3.67	1.97	0.049 *
GP Age	0.08	0.08	1.08	0.91	0.362
GP Sex—Female	0.03	1.00	1.03	0.03	0.973
GP CAKUT-Positive	−1.20	1.08	0.30	−1.11	0.266
GP Environment—Rural	−2.10	1.20	0.12	−1.75	0.080
GP Patient Effect	1.30	0.66	3.67	1.97	0.049 *

GNE—Gram-negative bacteria belonging to *Enterobacteriaceae* spp.; GNP—Gram-negative bacteria belonging to *Pseudomonaceae* spp.; GP—Gram-positive bacteria, * *p*-value < 0.05, statistically significant. In Bold—uropathogen group.

**Table 6 antibiotics-13-00684-t006:** Antibiotic resistance pathogens causing UTIs in children distributed in accordance with uropathogen type and correspondent antibiotic testing package.

Antibiotic	GNE Resistance	GNP Resistance	GP Resistance
Ampicillin	78.47%	100%	33.33%
Ampicillin/Sulbactam	85.71%	-	-
Amoxicillin	100%	-	-
Amoxicillin/Clavulanic Acid	30.19%	-	41.67%
Oxacillin	-	-	62.5%
Meropenem	5.77%	8.33%	0%
Imipenem	3.18%	16.67%	-
Ertapenem	1.97%	0%	
Piperacillin/Tazobactam	12.82%	8.33%	0%
Ticarcillin/Clavulanic Acid	-	12.5%	0%
Cefepime	20.86%	25%	-
Cefotaxime	34.65%	-	50%
Ceftazidime	29.81%	25%	50%
Ceftriaxone	1.3%	-	-
Gentamicin	34.38%	8.33%	18.18%
Amikacin	2.4%	8.33%	-
Ciprofloxacin	38.27%	25%	26.92%
Levofloxacin	2.06%	100%	-
Norfloxacin	50%		
Clindamycin	-	-	42.86%
Erythromycin	-	-	82.61%
Nitrofurantoin	18.59%	100%	-
Nalidixic acid	33.33%	-	30%
Fosfomycin	8.47%	0%	-
Trimethoprim/Sulphamethoxazole	61.49%	100%	-
Vancomycin	-	-	8.7%
Linezolid	-	-	4.17%
Colistin	50%	0%	-

GNE—Gram-negative bacteria belonging to *Enterobacteriaceae* Fam.; GNP—Gram-negative bacteria belonging to *Pseudomonaceae* spp.; GP—Gram-positive bacteria. The absence of value in table (-) signifies no pathogen testing for specific antibiotics in accordance with laboratory guidelines.

## Data Availability

The information is contained within this article in its entirety. For additional information, please feel free to inquire with either the original author or the corresponding author. The public’s access to the data is restricted as a result of the patient privacy standards that regulate the handling of clinical data.

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
