# Peer review of "General Characteristics and Current State of Antibiotic Resistance in Pediatric Urinary Tract Infection—A Single Center Experience"

_antibiotics, 2024, doi:10.3390/antibiotics13080684_

Round 1

Reviewer 1 Report

Comments and Suggestions for Authors

One of the main values of the study is that it brings a child population from a Central European country to be analyzed in terms of uropathogens.

The most valuable part of the work is Tables 4-6, where the reader can see the individual pathogens. The weakness of the work is the small number of elements and the fact that the data comes from a single place, and short study period.

I recommend the study for publication after the following minor corrections.

State the objective clearly in the abstract.

Please add, the application of a statistical test each of the  “p” values in the tables. (e.g. * = Fisher's exact test;  ** = X-squared test etc )

 In the case of bar chart, please mark the element numbers above each column/bar (Figures),  in addition to the textual legend, there should also be a graphic (Legend)

Figure 5 and Figure 6 Please sort the columns in descending order .

Table 5: Please enter the characteristic of the model. ()

Discussion: Ciprofloxacin and other quinolones cannot be given routinely in the pediatric population, please point out this in the discussion.

Limitations: another limitation of the study is the relatively short study period and the only one geographical location.

Author Response

Reviewer 1:

One of the main values of the study is that it brings a child population from a Central European country to be analyzed in terms of uropathogens.

The most valuable part of the work is Tables 4-6, where the reader can see the individual pathogens. The weakness of the work is the small number of elements and the fact that the data comes from a single place, and short study period.

Thank you very much for taking time and effort to read and review our manuscript and we are grateful for you positive and remarks and improvement suggestions.

I recommend the study for publication after the following minor corrections.

  1. State the objective clearly in the abstract.

Thank you for the positive correction. In the abstract we modified the aims of the study in accordance with your recommendations. The entire abstract was rephrased for a better understanding and emphasize of key findings. We added in the abstract of the manuscript details about the aim of the study: “The aim of the study is to bring a clear view of present-day particularities of UTI in children.”

We also rephrased the aims of the study in introduction section and added supplementary information. “The aim of the study is to bring a clear view of present-day particularities of UTI in children from west-part of Romania: global and CAKUT-related age/gender UTI distribution in children, UTI recurrence, diversity of uropathogens, pathogen - predictor variables (age, CAKUT status, environment, gender) correlations, general pediatric antibiotic resistance particularities and the effect of CAKUT presence upon antibiotic resistance.” -line 94-98

  1. Please add, the application of a statistical test each of the “p” values in the tables. (e.g. * = Fisher's exact test; ** = X-squared test etc )

Thank you for comment. We added the requested details. We added footnotes for Table 1 and 2 for each p-value to indicate the type of statistical test used. In Table 4 a header was added indicating the test that what used “Fisher’s Exact Test”

  1. In the case of bar chart, please mark the element numbers above each column/bar (Figures), in addition to the textual legend, there should also be a graphic (Legend).

Thank you for the suggestion. We have updated text as follows:

  • Figure 1: added numbers above each bar column, and a graphic legend;
  • Figure 2: added numbers above each bar column, a graphic legend and modified the labels to reflect our calculations
  • Figure 3: added numbers above each bar column, and a graphic legend;
  • Figure 4: added a graphic legend;
  • Figure 5: added numbers next to each bar column and added a graphic legend;
  • Figure 6: added numbers above each bar column, a graphic legend and ordered the bars in ascending order;
  1. Figure 5 and Figure 6 Please sort the columns in descending order.

Thank you for the valuable suggestion, we have combined Figures 5 and 6 and sorted the columns in ascending order. We also used green, orange and red colors to indicate the resistance status of the antibiotics included.

  1. Table 5: Please enter the characteristic of the model. ()

Thank you for the suggestion. For more clarity, we have added a short paragraph describing the mathematical form of the multinomial logistic regression model that was fit. Line 336-344.

  1. Discussion: Ciprofloxacin and other quinolones cannot be given routinely in the pediatric population, please point out this in the discussion.

Thank you for pointing out this aspect. We included additional information in the discussion section regarding fluoroquinolones use in pediatric patients. “The safety of Fluroquinolones use in pediatric population has been recurrently debated [41, 42, 43]. Ciprofloxacin, Levofloxacin, Moxifloxacin and Ofloxacin are approved for pediatric use with specific indications (complicated UTI/pyelonephritis, neonatal bacterial infections, multidrug-resistant gram-negative organisms) [41, 44]. This particular class of antibiotics is not routinely recommended in children as first-line treatment due to adverse reactions: musculoskeletal impairment, neuropathy, memory loss, hearing, taste, smell, vision impairment, QTc prolongation, etc.) [42, 45]. Musculoskeletal adverse effects caused by Ciprofloxacin have been reversible indicative of safe use in pediatric patients [46]. Line 650-658

  1. Limitations: another limitation of the study is the relatively short study period and the only one geographical location.

Thank you for your suggestion, we added the following details in the limitations of the study: “Also, the limited 6 months study period and single center experience and specific region of the country represent weaknesses of present study.” Line 715-717

Reviewer 2 Report

Comments and Suggestions for Authors

1. I would like to express my gratitude for the opportunity to review the manuscript entitled Updates on antibiotic resistance of uropathogens causing urinary tract infections in children in west-part of Romania” for possible publication in Antibiotics. However, there are major concerns that must be addressed in order to be considered for publication.

2. It is important for the author to address whether the sample size used in the study is representative of the west-part of Romania. Additionally, it is suggested that the author reconsider the research title to accurately reflect the study area.

3. There is a lack of comprehensive information on antimicrobial resistance and the analysis of resistance is not sufficiently in-depth. Additionally, I would suggest reconsidering the research title to better reflect the scope and generalizability of the study findings.

4. The conducting a Mann-Whitney U test to compare the ages of patients between different sex groups, as the age variable was not normally distributed. However, the author presented the mean age in Tables 1 and 2. It would be more appropriate to express the age variable using the median (IQR) instead of the mean (S.D.) to align with the non-normal distribution.

5. Including specific details about the disposable VITEK AST card in the manuscript would enhance transparency and reproducibility, allowing readers to understand the methodology and potentially replicate the study.

6. The patient age distribution (3 years old) with the presence or absence of malformation in Figure 2 seems to have an overlap between the age categories of 1-3 years and 3-6 years.

7. The author performed drug susceptibility analysis using VITEK® 2 Compact,but did not interpret or present the MIC values, such as MIC50, MIC90, and the ratio of MIC50/90. To enhance the comprehensiveness of the study results and facilitate a better understanding of the antimicrobial susceptibility profiles, it is recommended that the author include these important MIC values in the presentation of the findings.

8. Figures 5 and 6 present overlapping results; they are essentially the reverse of each other. It is recommended that the author avoids presenting redundant information and instead considers consolidating these findings into a single figure. Alternatively, the author could clearly differentiate the purpose and focus of each figure to minimize confusion for readers.

9. The author mentioned that there are significant differences in antibiotic resistance patterns between CAKUT and non-CAKUT groups. However, it seems that this comparison is based solely on the percentage of resistance in patients with and without malformation, rather than analyzing the actual resistance patterns. Could you please provide more comprehensive analysis of the resistance patterns in your study?

10. Please check for errors in the green square representing vancomycin in Figure 7.

11. Many of the results were presented in comparisons between males and females. However, a lack of hypothesis in the introductionand discussion sectionsregarding the importance and rationale behind considering gender as a factor of interest. 

12. The summary section of the manuscript should be both concise and comprehensive. It is essential to clearly state the key findings obtained from the substantive content and provide specific conclusions that offer valuable insights into clinical applications.

13. The use of “antimicrobial” and “antibiotic” is inconsistent throughout the text. It is important to clarify which words are most appropriate for this manuscript to ensure that the use of terminology is consistent and correct.

14. When mentioning a new species for the first time, it is important to write out the full genus name. However, in subsequent mentions, it is acceptable to use the abbreviated form of the genus. Please ensure to thoroughly review and verify the accuracy of the bacterial names throughout the entire manuscript.

Author Response

Reviewer 2:

  1. I would like to express my gratitude for the opportunity to review the manuscript entitled “Updates on antibiotic resistance of uropathogens causing urinary tract infections in children in west-part of Romania” for possible publication in Antibiotics. However, there are major concerns that must be addressed in order to be considered for publication.

Thank you very much for your time and implication for these very good suggestions. Your positive remarks and suggestions will definitely improve the manuscript in order to reach a publication standard.

  1. It is important for the author to address whether the sample size used in the study is representative of the west-part of Romania. Additionally, it is suggested that the author reconsider the research title to accurately reflect the study area.

Thank you for the observation. We initially failed to point the hospital`s important standing in the country`s region and its addressability, but hopefully with the newly added details the reader can have a clearer view. The present study was conducted in Emergency Hospital for Children `Louis Turcanu` Timisoara, which is the 1M regional pediatric hospital, the biggest hospital in the west part of Romania, with almost 40,000 Emergency Room presentations/year and 17,000 hospital admissions/year, responsible for approximatively one eight of pediatric population of the country. We added details regarding the hospital in the manuscript but better accuracy:

“The hospital is the biggest hospital in the west part of Romania, the 1M regional hospital, and is responsible for pediatric population in about one eighth of country`s area with almost 40,000 emergency room presentations/year and 17,000 hospital admissions/year.” Line103-106

We also reconsidered the title and changed it and we have confidence in the new one to accurately reflect the study area and research results.

  1. There is a lack of comprehensive information on antimicrobial resistance and the analysis of resistance is not sufficiently in-depth. Additionally, I would suggest reconsidering the research title to better reflect the scope and generalizability of the study findings.

Thank you for your comment. We decided to change the title, in accordance to your recommendations and to the content of the material into: “General characteristics and current state of antibiotic resistance in pediatric urinary tract infection – a single center experience”

  1. The conducting a Mann-Whitney U test to compare the ages of patients between different sex groups, as the age variable was not normally distributed. However, the author presented the mean age in Tables 1 and 2. It would be more appropriate to express the age variable using the median (IQR) instead of the mean (S.D.) to align with the non-normal distribution.

Thank you for the suggestion. We have updated the numbers in Table 1 and Table 2 such that the age and UTI count variables are summarized more appropriately using the median and IQR. We also applied changes to the text of the manuscript regarding median age and IQR.

  1. Including specific details about the disposable VITEK AST card in the manuscript would enhance transparency and reproducibility, allowing readers to understand the methodology and potentially replicate the study.

Thank you for the comment. Actually, we did not use disposable VITEK AST cards for antimicrobial susceptibility, I think we were a bit unclear in the explanations. We used VITEK 2 GP/GN identification cards for pathogen biochemical identification. We added more details in the manuscript for a clearer understanding:

“The bacterial strains isolated from each sample were identified by biochemical testing on VITEK® 2 Compact 15, VITEK 2 gram-positive (GP), and VITEK 2 gram-negative (GN) identification cards (bioMérieux, Marcy l'Etoile, France), following the manufacturer’s guidelines. For fungous, culture, direct microscopy and histopathology was used to identify pathogen.

Antibiotic sensibility was tested using the Kirby–Bauer disk diffusion method, according to CLSI (Clinical and Laboratory Standards Institute) guidelines. The bacterial suspensions were adjusted to 0.5McFarland standard solution and swabbed on the surface of Muller-Hinton agar (MHA) with a stick of 0.01ml with a standard calibrated loop and incubated at 35-37 Celsius degrees for 18-24 hours. The isolates were considered resistant, intermediate or susceptible to one antibiotic based on the diameter of the halo that appeared after the incubation period. The classification into the three bacterial groups (resistant, intermediate or susceptible) regarding antibiotic susceptibility was carried out according to the interpretation tables of CLSI guidelines [18].” Line 152-164

  1. The patient age distribution (3 years old) with the presence or absence of malformation in Figure 2 seems to have an overlap between the age categories of 1-3 years and 3-6 years.

Thank you for the comment. We made corrections to the Figure 2 description in accordance to statistical analysis datasheet.

  1. The author performed drug susceptibility analysis using VITEK® 2 Compact, but did not interpret or present the MIC values, such as MIC50, MIC90, and the ratio of MIC50/90.To enhance the comprehensiveness of the study results and facilitate a better understanding of the antimicrobial susceptibility profiles, it is recommended that the author include these important MIC values in the presentation of the findings.

Thank you for the positive remark. Our description of methods used was a little vague and maybe unclear. For antimicrobial susceptibility we used disk-diffusion technique, a qualitative method, in accordance to CLSI standards. For quantitative interpretation on VITEK 2 AST cards N222 and N204 could be used. Determining the MIC was not one of the objectives of current study, the purpose of the work being to bring a clear view of present-day particularities of UTI in children from west-part of Romania: global and CAKUT-related age/gender UTI distribution in children, UTI recurrence, diversity of uropathogens, pathogen - predictor variables (age, CAKUT status, environment, gender) correlations, general pediatric antibiotic resistance particularities and the effect of CAKUT presence upon antibiotic resistance. We hope we improved this section of the manuscript adequately enough for publication standards and also met your requirements. We added supplementary information and details to the document for a better understanding: “Antibiotic sensibility was tested using the Kirby–Bauer disk diffusion method, according to CLSI (Clinical and Laboratory Standards Institute) guidelines. The isolates were considered resistant, intermediate or susceptible to one antibiotic based on the diameter of the halo that appeared after the incubation period. The classification into the three bacterial groups (resistant, intermediate or susceptible) regarding antibiotic susceptibility was carried out according to the interpretation tables of CLSI guidelines [18].” Line152-164

  1. Figures 5 and 6 present overlapping results; they are essentially the reverse of each other. It is recommended that the author avoids presenting redundant information and instead considers consolidating these findings into a single figure. Alternatively, the author could clearly differentiate the purpose and focus of each figure to minimize confusion for readers.

Thank you for the valuable suggestion, we have combined Figures 5 and 6 and sorted the columns in ascending order. We also used green, orange and red colors to indicate the resistance status of the antibiotics included. The three groups of antibiotic resistance (High, intermediate and low) are also mentioned in Materials and Methods section.

  1. The author mentioned that there are significant differences in antibiotic resistance patterns between CAKUT and non-CAKUT groups. However, it seems that this comparison is based solely on the percentage of resistance in patients with and without malformation, rather than analyzing the actual resistance patterns. Could you please provide more comprehensive analysis of the resistance patterns in your study?

Thank you very much for the comment. Present study mentioned significant differences in antibiotic resistance between CAKUT and non-CAKUT groups, but not statistically significant due to the small sample size of CAKUT group (which has only 18 patients). We stated this aspect as limitation of the study. Unfortunately, in this manuscript the statistical analysis is delimited to comparison of resistance percentage between the two groups. This is certainly an interesting topic that we hope to develop in a future paper, once a more comprehensive data set is obtained. We added details in the manuscript regarding our study limitations and future perspectives and also softened the terms of “significant differences” in the paragraph of interest. Line 513-528

  1. Please check for errors in the green square representing vancomycin in Figure 7.

Thank you for your attentive comment. We have revised Figure 7 to show the size of the bar is 0%, since the pathogens did not show resistance for Vancomycin in any of the patients in the non-CAKUT group.

  1. Many of the results were presented in comparisons between males and females. However, a lack of hypothesis in the introduction, and discussion sections, regarding the importance and rationale behind considering gender as a factor of interest. 

Thank you for your comment and recommendations for improvement. We added details about gender discrepancies in children with UTI in hypothesis of the document, introduction section and discussion section:

In introduction section we added: “Gender represents a factor of interest when describing UTIs in children. Gender discrepancies are due to anatomical factors, presence of particular CAKUT in males (VUP) and genital sexual activity infections in adolescent girls [6] “. Line 60-62

The hypothesis was modified in accordance: “The aim of the study is to bring a clear view of present-day particularities of UTI in children from west-part of Romania: global and CAKUT-related age/gender UTI distribution in children, UTI recurrence, diversity of uropathogens, pathogen - predictor variables (age, CAKUT status, environment, gender) correlations, general pediatric antibiotic resistance particularities and the effect of CAKUT presence upon antibiotic resistance.” Line 94-98.

We added into the Discussion section: “In present study we investigated gender differences in children with UTI. 57% of patients were female, while 43% were males. The median age of hospital admission was 5 (IQR 12) years in females and a lower median in males of 2 years (IQR 7), statistically significant (p=0.03). No other significant differences were found between genders regarding environment origin, UTI count or presence/absence of CAKUT. Bimodal distribution reveals a high density of patients between 1-2 years of age for both sexes, but mainly young toddlers and small pediatric male patients develop UTI. Similarly, 13-15 years old adolescent females were identified to develop UTI with higher frequency. The EAU/ESPU guidelines reported an increased occurrence (20.3%) of UTIs in uncircumcised male infants and 5% in similar age female infants [21]. Male infants under the age of 6 months are approximately twice as likely to have an initial UTI rather than female same-age patients. Recurrence UTI risk in both males and females is equivalent in toddlers who have experienced an UTI (32% vs. 35% respectively) [22]. Also, it is documented that young females (12-14 years old) have a 3.2-fold risk of developing UTI compared to similar age males [23]. However, as children reach adolescent age (over 12 years), factors like sexual activity or hygiene practices influence the higher rate of UTIs in females [24]. Genger and presence/absence of CAKUT distribution showed a slightly higher occurrence of UTI for male toddlers associating CAKUT and for adolescent female patients, regardless of malformation. Our results were in accordance with literature findings [1, 2, 21, 23].” Line 535 - 554

  1. The summary section of the manuscript should be both concise and comprehensive. It is essential to clearly state the key findings obtained from the substantive content and provide specific conclusions that offer valuable insights into clinical applications.

Thank you for recommendation. We revised the entire abstract and included key findings in order to provide a better comprehending of important findings and highlight clinical applications, as follows:

“Urinary tract infection (UTI) represents one most common bacterial infection in children, mainly caused by gram-negative bacteria. Empirical antibiotic treatment is based on international and national guidelines for treating UTI in children and individualized with local antibiotic resistance patterns. The aim of the study is to bring a clear view of present-day particularities of UTI in children. Methods: We analyzed 210 positive urine cultures, identified in 141 pediatric patients admitted in hospital over a 6-months period. Results: The majority of patients were females (57%) with a median age of 5 years (IQR 12), while male patients revealed a median age of 2 (IQR 7). Most patients originated from urban areas (53%). Only 18 patients (12.76%) were identified with underlying Congenital Anomalies of Kidney and Urinary Tract (CAKUT). Escherichia Coli was the most frequent pathogen. Increased antibiotic resistance was found in commonly-used antibiotics Ampicillin and Trimethoprim/Sulfamethoxazole, and in case of patients with CAKUT. Suitable antibiotics in treating gram-negative UTI are aminoglycosides, Meropenem, IIIrd generation Cephalosporins and Nitrofurantoin. Vancomycin upholds efficacy in treating gram-positive pediatric UTI. Conclusion: Periodical analysis needs to be performed in order to constantly update clinicians on uropathogens` antibiotic resistance and optimal empirical treatment options.” Line 33-47

  1. The use of “antimicrobial” and “antibiotic” is inconsistent throughout the text. It is important to clarify which words are most appropriate for this manuscript to ensure that the use of terminology is consistent and correct.

Thank you for your comment. The most appropriate word for present study is antibiotic when referring to medicines used to prevent and treat bacterial infections. The definition of antibiotic resistance in accordance with WHO recommendations refers to bacteria`s change in response to antibiotics. Antimicrobial resistance is a broader term, encompassing resistance to drugs to treat infections caused by other microbes as well, such as parasites (e.g. malaria), viruses (e.g. HIV) and fungi (e.g. Candida). We have made corrections throughout the document in accordance with topic. The subtitle from Materials and methods („Antimicrobial susceptibility”) remained unchanged as we talk about both bacteria and fungous identification.

  1. When mentioning a new species for the first time, it is important to write out the full genus name. However, in subsequent mentions, it is acceptable to use the abbreviated form of the genus. Please ensure to thoroughly review and verify the accuracy of the bacterial names throughout the entire manuscript.

Thank you for the positive remark and correction. We corrected within the document, the nomenclature of bacteria accordingly.

Also, the reference list will be modified due to the reference inclusion.

Round 2

Reviewer 2 Report

Comments and Suggestions for Authors

Thank you for your timely and comprehensive response to my comments on your manuscript. After carefully reviewing your responses, I am delighted to inform you that I wholeheartedly accept the manuscript in its current form.